# SIRT1/FOXO Signaling Pathway in Breast Cancer Progression and Metastasis

**DOI:** 10.3390/ijms231810227

**Published:** 2022-09-06

**Authors:** Sayra Dilmac, Nilay Kuscu, Ayse Caner, Sendegul Yildirim, Burcak Yoldas, Ammad Ahmad Farooqi, Gamze Tanriover

**Affiliations:** 1Department of Histology and Embryology, School of Medicine, Akdeniz University, 07070 Antalya, Turkey; 2Department of Nanomedicine, Houston Methodist Research Institute, Houston, TX 77060, USA; 3Cancer Research Center, Ege University, 35040 Izmir, Turkey; 4Department of Parasitology, School of Medicine, Ege University, 35040 Izmir, Turkey; 5Department of Experimental Therapeutics, MD Anderson Cancer Center, Houston, TX 77030, USA; 6Department of Medical Biology and Genetic, School of Medicine, Akdeniz University, 07070 Antalya, Turkey; 7Department of Molecular Oncology, Institute of Biomedical and Genetic Engineering, Islamabad 54000, Pakistan; 8Department of Medical Biotechnology, School of Medicine, Akdeniz University, 07070 Antalya, Turkey

**Keywords:** SIRT1, FoxO, primary tumor, metastasis, breast cancer

## Abstract

Breast cancer is the second most common cancer in women. The roles of the SIRT and FoxO proteins in tumor progression are known, but their roles in metastasis have not yet been clearly elucidated. In our study, we investigated the roles of SIRT and FoxO proteins their downstream pathways, proteins p21 and p53, in tumor progression and metastasis. We evaluated these proteins in vitro using metastatic 4TLM and 67NR cell lines, as well as their expression levels in tumor-bearing mice. In addition, the regulatory role of SIRT and FoxO proteins in different transduction cascades was examined by IPA core analysis, and clinicopathological evidence was investigated in the TCGA database. In primary tumors, the expression levels of SIRT1, p21, p53, E2F1 and FoxO proteins were higher in 67NR groups. In metastatic tissues, the expression levels of SIRT1, E2F1 and FoxO proteins were found to be enhanced, whereas the levels of p53 and p21 expression were noted to be reduced. IPA analysis also provided empirical evidence of the mechanistic involvement of SIRT and FoxO proteins in tumor progression and metastasis. In conclusion, SIRT1 was found to co-operate with FoxO proteins and to play a critical role in metastasis. Additional research is required to determine why overexpression of SIRT1 in metastatic tissues has oncogenic effects.

## 1. Introduction

Breast cancer is the most frequent type of cancer diagnosed in women and the most common cause of cancer-related death worldwide [1,2]. The overall average 5-year survival rate for patients with early-stage breast cancer is higher than that of patients diagnosed in later stages. The main cause of breast-cancer-related deaths is not primary tumors but metastatic spread to distant organs [3]. Although metastasis is the major cause of treatment failure in cancer patients, metastasis-associated mechanisms have not yet been fully clarified.

Proto-oncogenes are involved in the regulation of the cell cycle, but their overexpression can also lead to uncontrolled cell division [4]. Tumor suppressor genes (TSGs) are known to negatively regulate tumor growth by controlling cell division; however, their downregulation can also increase proliferation and the metastatic ability of cancerous cells [5]. Mutation of either proto-oncogenes or TSGs can trigger tumorigenesis and cancer metastasis. As proto-oncogenes and TSGs are crucial, it is important to understand their mechanistic roles and those of their related pathways in tumorigenesis and metastasis.

Sirtuins (SIRTs) (silent information regulators) are NAD (Nicotinamide adenine dinucleotide)-positive–dependent class III histone deacetylases (HDACs) [6] are a novel oncogene family [7]. SIRT1 plays an important role in cell survival by regulating the transcriptional activities of p53, inducing apoptosis [8,9] and suppressing FoxO proteins [10,11]. SIRT1 overexpression has been shown to cause tumor growth and a significant increase in the cell survival ability of cancer cells [9].

The forkhead box O (FoxO) family of transcriptional factors comprises of four members: FoxO1, FoxO3, FoxO4 and FoxO6 [12]. FoxO transcription factors are involved in various crucial mechanisms, such as apoptosis, cell cycle arrest, resistance to oxidative stress, the DNA (deoxyribonucleic acid) repair mechanism, glucose metabolism, energy, homeostasis and cellular differentiation [13]. Moreover, FoxO proteins are “double-edged swords” dualistically involved in regulation of various steps of carcinogenesis and metastasis [12]. Oxidative stress causes the transactivation of FoxO by catalyzing its deacetylation in an NAD-dependent manner regulated by SIRT1 [14]. Under stress conditions, such as apoptosis and DNA repair regulation, SIRT1 forms a complex in the nucleus, deacetylates the FoxO proteins and affects FoxO1 by reducing stress [10]. If FoxO is inhibited, the gene transcription required in the apoptosis steps cannot be induced [15]. Taken together, these findings suggest that the interaction between SIRT1 and FoxO1 may play a role in the metastasis process and the treatment of breast cancer. Hu Q. et al. showed that inhibition of SIRT1 increased FoxO3a and acetylated FoxO3a in bladder cancer cells. Increased FoxO3a acetylation was reported to affect cell cycle regulation and antioxidant response [16]. However, the roles of SIRT1 and FoxOs in breast cancer metastasis have not been elucidated to date.

Mutations in the p53 gene, which regulates cell cycle and apoptosis, causes loss of tumor-suppressor functions, resulting in tumor progression [17]. The SIRT1–p53 axis plays a complex role in tumorigenesis, with dual functions in tumor promotion and tumor suppression [18,19]. p21 (CDK1, cyclin-dependent kinase 1), also known a cyclin-dependent kinase inhibitor [20], is responsible for the regulation of the cell cycle in the G1 and S phases, and its expression is controlled by the p53 protein [21]. FoxOs and p53 share several downstream target genes, such as p21 [22], suggesting that FoxOs and p53 may also coregulate tumor-suppressor signaling.

E2F (E2F transcription factor), comprising eight genes, plays critical roles in cell cycle regulation [23]. E2F1 (E2F transcription factor 1) promotes p53-mediated apoptosis by inducing the expressions of apoptosis-related proteins [24,25]. In addition, decreased expression of E2F1 and apoptosis-related genes has been shown to adversely affect patient survival in breast and ovarian cancer patients [26]. Considering these studies, E2F1 appears to play a bidirectional role in cell survival.

Based on findings regarding the roles of SIRT1 and FoxOs in tumor growth and apoptosis, as well as their roles on metastasis, the aim of the present study was to investigate expression and regulation of SIRT1 and FoxO and their related pathways in tumor growth and metastasis in primary tumors and distant organs using both benign and highly metastatic breast cancer cells under in vitro and in vivo conditions. First, we sought to clarify the dual role of SIRT1 as both a tumor suppressor and a promoter of tumor growth; secondly, we investigated the role of FoxO proteins during tumor progression and metastasis.

## 2. Results

### 2.1. In Primary Tumors, Both 67NR and 4TLM Tumor Cells Exhibited Differential Expression Levels and Localizations of SIRT1 and FoxOs

The expressions of SIRT and FoxO proteins, as well as p53 and p21 proteins, were determined in both benign (67NR) and malignant (4TLM) breast cancer tumor cells. The expression of SIRT1 protein was limited to the nuclear level and higher in 67NR cells compared to 4TLM (Figure 1a). The expression of p53 was cytoplasmic in 67NR cells, whereas it was nuclear in 4TLM cells (Figure 1a). The expression of p21 was localized in the nucleus in both 67NR and 4TLM cells but less so in 4TLM compared to 67NR cells (Figure 1a). FoxO1, FoxO3a and FoxO4 exhibited distinct cytoplasmic expressions in 4TLM and 67NR cell lines (Figure 1b). FoxO1 expression was higher in 4TLM cells compared to 67NR cells, and the expression FoxO3 and FoxO4 was higher in 67NR cells compared to 4TLM cells.

### 2.2. Expression of SIRT1 and FoxOs Differed between 67NR and 4TLM Primary Tumors in Mice

Expression levels of SIRT1 and FoxO proteins in primary tumors were evaluated in vivo to reveal the differences between metastatic and non-metastatic tumors. In addition, p53, p21, E2F1 and cleaved caspase 3 proteins were investigated to reveal the relationship between the metastatic potential of tumors and cell viability. SIRT1 protein is expressed in both the cytoplasm and nucleus in primary tumors. However, the expression level of SIRT1 was significantly decreased in metastatic 4TLM compared to non-metastatic 67NR cells (*p* ˂ 0.05) (Figure 2a,b).

FoxO1 was expressed in the cytoplasm in primary tumors, and its expression level was significantly decreased in metastatic 4TLM compared to non-metastatic 67NR cells (*p* ˂ 0.05). FoxO3a was expressed in both the cytoplasm and nucleus of non-metastatic 67NR cells, whereas it was expressed only in the cytoplasm of metastatic 4TLM cells. FoxO3a expression was significantly decreased in metastatic 4TLM compared to non-metastatic 67NR cells (*p* ˂ 0.05). FoxO4 was expressed in the cytoplasm in primary tumors, and its expression level was significantly decreased in metastatic 4TLM compared to non-metastatic 67NR cells (Figure 2c,d) (*p* ˂ 0.05).

In addition, both p53 and p21 proteins are expressed in the cytoplasm in primary tumors, and their expressions levels were significantly decreased in metastatic 4TLM compared to non-metastatic 67NR cells (*p* ˂ 0.05). In the 67NR primary tumors, E2F1 was expressed both in the nucleus and the perinuclear region, and its expressions was higher compared to that in 4TLM cells (*p* ˂ 0.05). However, E2F1 expression was limited in the perinuclear region and lower in 4TLM than 67NR tumors (Figure 2a,b). Nuclear cleaved caspase 3 expression was increased in metastatic 4TLM primary tumors compared to non-metastatic 67NR tumors (Figure 2c,d) (*p* ˂ 0.05).

Moreover, the gene expression profiles of SIRT1, FoxO1, FoxO3a and FoxO4 in primary tumors were investigated by RT-qPCR analysis. The mRNA (messenger ribonucleic acid) expression levels of SIRT1, FoxO1 and FoxO3a were significantly decreased in 4TLM metastatic tumors compared to 67NR non-metastatic tumors, whereas the FoxO4 signal was significantly increased in metastatic tumors (Figure 2e) (*p* ˂ 0.05).

### 2.3. Expression Levels of SIRT1 and FoxOs in Metastatic Tissues

The expression levels of SIRT1 and FoxO proteins, as well as those of p53, p21, E2F1 and cleaved caspase 3 proteins, were evaluated in metastatic liver and lung tissues of mice. SIRT1 is expressed in the cytoplasm and nucleus in liver tissues, and its expression levels increased in metastatic 4TLM and non-metastatic 67NR liver tissues compared to tumor-free liver tissue (*p* ˂ 0.05). Moreover, SIRT1 expression was higher in metastatic areas in metastatic 4TLM compared to 67NR tumors. SIRT1 expression was observed in infiltrative cells of tumors around the hepatic artery in metastatic microenvironments (Figure 3a,b). Cytoplasmic p53 expression increased only the 4TLM group compared to tumor-free liver tissue (*p* ˂ 0.05). Similarly to SIRT1, nuclear p53 expression was observed in infiltrative cells from tumors around the hepatic artery (Figure 3a). p21 was expressed in the cytoplasm of liver cells, with no significant differences observed among three groups. p21 was expressed specifically in Kupffer cells in tumor-free liver tissue (Figure 3a). In addition, E2F1 was expressed in the perinuclear region of hepatocytes in the 4TLM and 67NR groups. Cytoplasmic E2F1 expression was observed in infiltrative cells from tumors and Kupffer cells in the 4TLM group. E2F1 expression was lower in the tumor-free liver group and significantly increased in the 4TLM and 67NR groups compared to the tumor-free group (Figure 3a). There was no significant difference observed between the 4TLM and 67NR groups in terms of E2F1 expression (*p* ˂ 0.05) (Figure 3a,b).

FoxO1was expressed in the cytoplasm in liver tissues in metastatic 4TLM non-vascular immune cells (*p* ˂ 0.05), and its expression was not determined in non-metastatic 67NR and tumor-free liver tissue (Figure 3c,d). FoxO3a was expressed in the cytoplasm in all liver tissues, and its expression levels increased only in metastatic 4TLM non-vascular immune cells compared to the tumor-free group (*p* ˂ 0.05). FoxO4 was expressed in the cytoplasm in liver tissues in the 4TLM and 67NR groups, and its expressions levels were higher in both metastatic 4TLM and non-metastatic 67NR liver tissue non-vascular immune cells (*p* ˂ 0.05) compared to tumor-free liver tissue, in which FoxO4 expression was not determined (Figure 3c). Although FoxO1, FoxO3a and FoxO4 were expressed in infiltrative cells from tumors around the hepatic artery, they were not expressed in hepatocytes (Figure 3d). Cleaved caspase 3 expression was not determined in any of the groups (Figure 3c).

In lung tissue, cytoplasmic expression of SIRT1 was more extensively expressed in metastatic areas in the metastatic 4TLM compared to non-metastatic 67NR and tumor-free groups. SIRT1 expression was increased significantly in metastatic lesions in 4TLM lung tissue (Figure 4a,b) (*p* ˂ 0.05). Cytoplasmic p53 expression was observed in alveolar cells in the tumor-free and 67NR groups but not in metastatic areas in the 4TLM group (Figure 4a). There was no significant difference between groups in terms of p53 expression (Figure 4b). Cytoplasmic p21 expression was lower in the metastatic 4TLM and non-metastatic 67NR groups compared to the tumor-free group (Figure 4b) (*p* ˂ 0.05). Although p21 expression was lowest in the 4TLM group, p21 was strongly expressed in metastatic areas (Figure 4a). Cytoplasmic expression of E2F1 in lung tissue was higher in the non-metastatic 67NR and metastatic 4TLM groups compared to the tumor-free group and was strongly expressed in metastatic areas in the 4TLM group (Figure 4a,b) (*p* ˂ 0.05).

FoxO1 expression was limited only in the cytoplasm of metastatic cells in the 4TLM group, and its expression was not determined in the 67NR and tumor-free groups (Figure 4c,d) (*p* ˂ 0.05). Cytoplasmic FoxO3a and FoxO4 proteins were expressed in metastatic cells in the 4TLM group, similarly to FoxO1, and their expression levels were higher in the 4TLM compared to 67NR and tumor-free groups (Figure 4d) (*p* ˂ 0.05).

### 2.4. Functional Enrichment Analysis

As an elementary investigation of the molecular mechanisms related to SIRT1/FoxO underlying breast cancer, microarray data were submitted to IPA (ingenuity pathway analysis). Differentially expressed genes (FDR < 0.05 and absolute log2 FC > 1) were categorized into related canonical pathways based on the ingenuity pathway knowledge base. For DEGs (differentially expressed genes) mapped to IPA (genes not mapped to the IPA database were excluded in our pathway analysis), 113 significant canonical pathways were identified in three groups (BH-adjusted *p*-value < 0.01). The most enriched categories of canonical pathways with absolute *p*-value and z scores of more than 1.3 and 2, respectively, in primary breast cancer, liver and lung metastasis are presented and compared in Figure 5. Calcium signaling was significantly increased in metastasis, whereas Th1 (T helper type 1) signaling was significantly decreased compared to primary breast cancer. Hepatic fibrosis signaling and inflammation signaling had the highest activation scores in metastatic liver tissue. The sirtuin signaling pathway, which is modulated in both primary tumors and metastasis by the FoxO and SIRT gene families, was activated in the primary tumors, liver and lung metastasis groups (Figure 5a–c).

Upstream regulator analysis is a novel function available in IPA by analyzing linkage to DEGs through coordinated expression to identify potential upstream regulators, including transcription factors and genes, which have been experimentally observed to affect gene expression. It was recently used to robustly identify the FoxO family as an important regulator in breast cancer and metastasis [27]. FoxO1 (*p* = 1.19 × 10^−4^, 5.16 × 10^−7^), FoxO3 (*p* = 2.94 × 10^−5^, 3.07 × 10^−7^) and FoxO4 (*p* = 2.39 × 10^−8^, 1.0 × 10^−6^) were predicted to be upstream regulators in liver and lung metastasis, whereas only FoxO1 (*p* = 2.6 × 10^−3^) and FoxO4 (*p* = 4.59 × 10^−2^) were predicted as upstream regulators in primary breast tumors. Furthermore, SIRT1 (*p* = 4.46 × 10^−5^) in primary tumors, SIRT1 and SIRT6 (*p* = 3.4 × 10^−4^, 1.8 × 10^−1^) in liver metastasis and SIRT1 and SIRT2 (*p* = 3.27 × 10^−9^, 5.17 × 10^−3^) in lung metastasis played roles as upstream regulators. The target genes affected by FoxO1, FoxO3, FoxO4 and SIRT1 in liver and lung metastasis are listed in Table 1. Regulator effects elucidated through IPA explain how predicted activated or inhibited upstream regulators might cause increases or decreases in downstream phenotypic or functional outcomes. Cellular proliferation of tumor cells was found to have a regulator effect in metastases of 4T1 cells (Figure 5d).

### 2.5. Clinicopathological Statistics of TNBC Patients 

Analysis of the mRNA expression of the FoxO family and SIRT1 in TCGA (The Cancer Genome Atlas) revealed that all genes were downregulation in tumors compared to normal tissues in TNBC (triple-negative breast cancer) (*p* < 0.05 for FoxO1, FoxO4) (Figure 6a,b). However, FoxO1, FoxO4 and SIRT1 were upregulated in the metastasis stage of TNBC, although the number of patients in the sample was too low to evaluate (Figure 6c) (*p* > 0.05).

## 3. Discussion

The roles of the SIRT1/FoxOs pathway in breast cancer and its metastasis have not been explained to date. SIRT1 is known to regulate oncogenic signals and play a role in the formation of the appropriate microenvironment for tumor cell survival [28]. FoxO transcription factors are deacetylated by SIRT1 [14], and deacetylated FoxO transcription factors regulate cellular signals, such as apoptosis, DNA damage, and cell survival [10,15,29]. The role of SIRT1 and FoxOs as regulators in important signaling pathways in tumor progression suggests that they are closely related to cancer.

It is known that increased activity of SIRT1 inhibits p53 by deacetylation. On the other hand, when SIRT1 activates p53, cancer formation is prevented [28,30]. Our results show that both 4TLM and 67NR cells express SIRT1 and p53. SIRT1 expression was limited in the nucleus in 67NR cells. It is known that nuclear translocation is prevented by SIRT1 deacetylation; however, it increases p53 accumulation in the cytoplasm [30]. According to our results, p53 is expressed in the cytoplasm, and SIRT1 is expressed in the nucleus, which is consistent literature reports. Similarly, p21 expression in the cytoplasm is known to arrest the cell cycle [31]. In tumor cells, p21 and p53 are localized in the cytoplasm under in vitro conditions. In addition, more intensive expression of p21 and p53 in the cytoplasm of non-metastatic tumor cells compared to metastatic tumor cells showed that 4TLM cells are more proliferative because the cell cycle continues rapidly in 4TLM cells. According to Western blot results in primary tumors, the expression of SIRT1, p21, p53, E2F1 and FoxO proteins was also higher in the 67NR group in primary tumors, similar to our immunohistochemistry results. Increased expression of SIRT1, FoxO1 and FoxO3a in the 67NR group confirms our results which related to the metastatic character of the tumor.

Inhibition of SIRT1 is known to increase apoptosis by causing p53 activation, in addition to reducing tumor growth [32]. According to our results, SIRT1 expression was higher in 67NR compared to 4TLM cells. These results revealed that tumor suppressor activity of SIRT1 is active in non-metastatic tissue. On the other hand, low expression of p53 and p21 may indicate that 4TLM tumor cells could escape from apoptosis. It has been shown that SIRT1 gene expression was high in the 67NR group, which may be associated with the tumor-suppressor effect of SIRT1 [32]. In our study, similar results were supported by immunohistochemistry, Western blot and PCR analysis. Guttila I.K. et al. showed that FoxO1 gene expression was higher in normal breast tissue than in breast tumor tissue [33].

E2F1 is a key regulator of the cell cycle [23,34]. Mori K. et al. reduced cell proliferation by inhibition of E2F1, which is a key regulator of the cell cycle in breast cancer cells [35]. According to our results, perinuclear localization of E2F1 in 4TLM primary tumors may reduce cell proliferation. Our results with respect to p53, p21 and E2F1 indicate that metastatic 4TLM cells could escape from apoptosis and continue their tumor progression. In contrast, cleaved caspase 3 expression was high in 4TLM primary tumors, particularly in necrotic areas.

SIRT1 is known to be associated with metastasis, with a role in cell proliferation and tumor development. Jin X. et al. showed that metastasis is triggered in breast cancer when SIRT1 expression is increased by lentivirus. High expression levels of SIRT1 cause increased invasion in breast cancer cells, whereas SIRT1 inhibition by shRNA decreases the invasion of breast cancer cells. This evidence suggests that SIRT1 is associated with metastasis [36]. According to our results, the highest SIRT1 expression in 4TLM cells in liver and lung tissue is related to SIRT1-triggered invasion. SIRT1 expression was very low in the 67NR groups and in the tumor-free groups. We also observed that the level of SIRT1 expression was increased in metastasis of TNBC patients (Figure 6c), whereas it was downregulated in primary tumors of TNBC patients compared to normal tissue (Figure 6a). Taken together, our results suggest that SIRT1 could play an important role in metastasis, in addition to other roles.

FoxO transcription factors are known to function as tumor suppressors in cancers [27]. Localization of the FoxOs determine their activity status. FoxOs must be expressed in the nucleus to activate their target genes, which are related to crucial cellular processes. When FoxOs are translocated from the nucleus to the cytoplasm by growth factors, they cannot function, resulting in inhibition of their tumor-suppressor function [37]. These results suggest that tumor-suppressor FoxO proteins are retained in the cytoplasm, and tumor cells are preserved. Several studies shown the FoxOs play a supportive role in facilitating and even stimulating metastasis [38,39,40]. Whereas increased FoxO3a expression reduces metastasis, it is does not affect primary tumor growth [38]. In our study, in primary tumors, FoxO3a expression was increased in non-metastatic 67NR compared to 4TLM cells (Figure 2d). In addition, FoxO1 and FoxO4 expression in metastatic 4TLM primary tumors was lower than in non-metastatic primary tumors. Similarly, the level of FoxO3 expression was decreased in metastasis of TNBC patients, whereas the expression of FoxO1 and FoxO4 was increased (Figure 6c). Limited expression of FoxOs in metastatic tumors is thought to be closely related to the metastatic character of the tumor. Moreover, our results show that FoxO expression was significantly higher in the 4TLM group compared to the non-metastatic and control groups in the liver, which is a metastatic organ. The expression of FoxO1, FoxO3a and FoxO4 proteins was higher in 4TLM lung tissue, especially in metastatic cells. Additionally, our transcriptomic analysis revealed the same results in both liver and lung metastases of mice (Figure 5c,d).

According to our results, FoxO1 expression was increased in 67NR compared to 4TLM cells. Liu H. et al. showed that reduced expression of FoxO3a promotes tumor progression by supporting stem cell characters of tumor cells. FoxO3a expression was downregulated in metastatic breast cancer tissues compared to normal breast epithelial cells (MCF-10A). We also observed that FoxO genes were downregulated in TNBC patients (Figure 6a). In addition, the downregulation of FoxO3 leads to changes in the levels of CD44/CD24, which are breast cancer stem cell markers [41]. In our study, we showed that FoxO3a expression was low in primary tumors obtained from 4TLM groups, which are highly metastatic. This result supports the hypothesis that FoxO3a affects the metastatic character of tumors. In prostate cancer, FoxO3a mRNA expression was increased in high-grade tumor samples compared to benign prostate cancer tissue [42]. We observed that FoxO4 mRNA levels were increased in the metastatic 4TLM group compared to the non-metastatic 67NR group. However, FoxO4 protein levels decreased in 4TLM metastatic primary tumors.

Analysis of transcriptomic data revealed that some pathways are shared among primary tumors, metastatic lung and liver tissue, despite differences in transcript abundance (Figure 5). Calcium signaling was enriched in metastases, whereas the Th1 signal was more activated in primary tumors. The potential role of T cells in promoting one of the most important steps in metastasis was previously established [43]. Th1 cells dramatically decrease the incidence of metastases without altering the development of primary tumors [44]. The contribution of the calcium signal to metastasis of cancer cells has been reviewed in previous studies [45]. Although physiological levels of Ca^2+^ inhibit proliferation and invasion, high Ca^2+^ levels ultimately increase the risk of metastasis in breast cancer [46]. Moreover, hepatic fibrosis signals, inflammation signals and the adipose tissue pathway were markedly more activated in metastatic liver tissue than other tissues. According to canonical pathway analysis, the SIRT signaling pathway, which consists of SIRT family members and FoxOs, was activated in both lung and liver metastasis (Figure 5a). The SIRT pathway plays a key role in the regulation of genes involved in the metastatic processes in various cancers, including breast cancer [47]. This pathway shows how cell proliferation and tumor growth are induced in lung and liver metastases, in line with our results (Figure 5b).

Furthermore, upstream regulator analysis by IPA identified SIRT1, FoxO1, FoxO3 and FoxO4 as upstream regulators in lung and liver metastasis (Table 1). Yadav R. K. et al. reported that FoxOs generally serve as a central regulator of cellular homeostasis and cancer metabolism and are tumor suppressors in many human primer cancers [27]. However, we observed that FoxOs genes regulated multiple involved in the invasion, cell cycle, proliferation, apoptosis, ROS (reactive oxygen species) production, inflammation and adipogenesis in metastases, in concurrence with the experimental data obtained in the present study.

In conclusion, our results show that the expressions of FoxO proteins in primary tumors was markedly lower in 4TLM groups compared to the 67NR group. Decreased expression of FoxO proteins suggests that they could induce metastasis. The expression observed in metastatic liver tissues suggests that SIRT1 can deacetylate FoxO proteins by forming a complex in the nucleus with FoxO [10]. The expression of SIRT1 and FoxO proteins in the lungs and liver, especially in metastatic cells, indicates that they may play crucial roles in metastasis [36,38]. In addition, through IPA analysis and TCGA data, we showed that FoxO1, FoxO3, FoxO4 and SIRT1 are associated with primary tumors and metastasis. For the first time in the literature, our results show that SIRT1 and FoxO proteins are associated with metastasis of breast cancer. Moreover, our findings could lead to further studies related to signaling mechanisms that still remain to be investigated with respect to whether SIRT1 and FoxO proteins support metastasis.

## 4. Materials and Methods

### 4.1. In Vitro Experimental Procedures

#### 4.1.1. Cell Culture

4T1 cells were previously derived from spontaneously formed breast tumors in BALB/c mice, as previously described [48]. The 4THM (4T1-Heart Metastasis) cell line was derived from cardiac metastasis of 4T1 cells, as described by Erin N. et al. [49]. These 4THM cells were implanted orthotopically into BALB/c mice, thereby establishing macroscopic liver metastasis, which was used to develop an additional cell line designated 4TLM (4T1-liver metastasis) [50]. We also used the 67NR mouse cell line as a non-metastatic breast cancer. 67NR and 4TLM cell lines were grown in DMEM-F12 (Dulbecco’s Modified Eagle Medium/Nutrient Mixture F-12) (Invitrogen; #11320074, Waltham, MA, USA) supplemented with 5% FBS (fetal bovine serum) (Invitrogen; #10270106, Waltham, MA, USA), 2 mM L-glutamine (Invitrogen; #25030024, Waltham, MA, USA), 1 mM sodium pyruvate (Invitrogen; #11360039, Waltham, MA, USA) and 0.02 mM non-essential amino acids (Invitrogen; #11140035, Waltham, MA, USA).

#### 4.1.2. Immunocytochemistry

The expression of SIRT1, p53, p21 and FoxO proteins in 4TLM and 67NR cells was evaluated using immunocytochemistry. The cells were fixed in 4% paraformaldehyde (Merck; 1.04005.1000, Rahway, NJ, USA) for 5 min and washed in phosphate-buffered saline (PBS). Then, the cells were blocked with blocking solution containing 1.5 g bovine serum albumin (Euroclone; EMR086025, Pero, Milan, Italy) and 0.0375 glycine (Bio-Rad; #161-0718, Hercules, CA, USA) for 15 min and incubated with anti-SIRT1 (Santa Cruz; #sc-15404, CA, USA, 1/100 dilution), anti-p53 (Santa Cruz; #sc-6243, Dallas, TX, USA 1/100 dilution), anti-p21 (Santa Cruz; #sc-756, Dallas, TX, USA 1/100 dilution), anti-FoxO1 (Cell Signaling; #2880S, Danvers, MA, USA 1/100 dilution), anti-FoxO3a (Cell Signaling; #12829S, MA, USA 1/100 dilution) and anti-FoxO4 (Santa Cruz; #sc-25539, CA, USA 1/100 dilution) for 2 h at room temperature. After washing out the primary antibodies, cells were incubated with the Alexa Fluor 555 (Thermo Fisher; #A-31572, Waltham, MA, USA 1/250 dilution) and Alexa Fluor 488 secondary antibody (Thermo Fisher; #A-21206, MA, USA 1/250 dilution) for 1 h at room temperature. The mounting medium containing DAPI (4′,6-diamidino-2-phenylindole) (Vector Labs; #H-1200, Newark, CA, USA) was dropped, and the expression was evaluated and photographed under an Olympus BX61 fluorescence microscope (Carl Zeiss GmbH, Jena, Germany).

### 4.2. In Vivo Animal Studies

#### 4.2.1. Animal Models

Female BALB/c mice were obtained from Kobay Animal Laboratory (Ankara, Turkey), kept under a 12 h light–dark cycle and fed a controlled diet. All experimental protocols were approved by the Local Ethics Committee for Animal Research (2 September 2016). 4TLM (1 × 10^5^) and 67NR (1 × 10^6^) cells in HBSS (Hanks’ Balanced Salt Solution) (Sigma Aldrich, St. Louis, MI, USA; #H9269, GE) were injected into the right upper mammary fat pad just beneath the armpit of BALB/c mice under ketamine/xylazine anesthesia (15 mg/kg i.m.). The formation of 67NR tumors requires implantation of a higher number of cells. Eight animals were used for each group. Breast cancer can easily spread to other parts of the body, most commonly to the bones, lungs or liver [51,52]. Therefore, to evaluate the expression of SIRT1 and FoxO in metastatic tissues, we selected lungs and livers, where breast cancer metastasis is common, and collected these tissues, along with primary tumors.

#### 4.2.2. Immunohistochemistry

Primary breast tumors, as well as lung and liver tissues, were removed 27 days after injection of tumor cells. All tissues were fixed in 10% formaldehyde (Merck; 1.04003.1000, Rahway, NJ, USA) for 24 h, dehydrated through a graded ethanol series and embedded in paraffin. Five μm thick sections were taken. After deparaffinization and rehydration, citrate buffer (pH 6.0) was used for antigen retrieval under microwave. Then, the slides were washed, and endogenous peroxidase activity was blocked by 3% hydrogen peroxide (Merck; #1.08600.1000, MA, USA) in methanol (Merck; #1.060.092.511, MA, USA) for 15 min at room temperature. After washing in PBS, universal blocking reagent (Thermo Fisher; #TA-125-UB, Waltham, MA, USA) was used to block the nonspecific bindings for 7 min at RT. Then, the sections were incubated with SIRT1 (Santa Cruz; #sc-15404, Dallas, TX, USA 1:200 dilution), p21 (Santa Cruz; #sc-756, Dallas, TX, USA, 1:200 dilution), p53 (Santa Cruz; #sc-6243, Dallas, TX, USA 1:200 dilution), E2F1 (Abcam; #ab-179445, Cambridge, UK 1:100 dilution), FoxO1 (Cell Signaling; #2880S, Danvers, MA, USA, 1:100 dilution), FoxO3a (Cell Signaling; #21829S, Danvers, MA, USA, 1:100 dilution), FoxO4 (Cell Signaling; #2359, Danvers, MA, USA, 1:100 dilution) and cleaved caspase 3 (Cell signaling; #9661L, Dancers, MA, USA, 1:50 dilution) antibodies overnight at +4 °C. After washing several times in PBS, sections were incubated with biotinylated goat anti-rabbit IgG (Immunoglobulin G) secondary antibody (Vector Lab; #BA1000, Newark, CA, USA, 1:400 dilution) for 1h at room temperature, followed by incubation with HRP (Horseradish peroxidase) streptavidin-peroxidase complex (Invitrogen; #85–9043, Waltham, MA, USA) for 20 min at room temperature. All incubation steps were performed in a humidified chamber to avoid dehydration of the slides.

Positive immunoreactions were visualized by incubation of 3,3′-diaminobenzidine (DAB) (Sigma Aldrich; #D4168, St. Louis, MO, USA) chromogen and counterstained with Mayer’s hematoxylin (Merck; #1.09249.1000, Rahway, NJ, USA). All expressions were evaluated and photographed under a Zeiss-Axioplan microscope (Carl Zeiss GmbH, Jena, Germany).

### 4.3. Image J Analysis of Immunohistochemical Staining

Micrographs of tissue samples from all groups were taken using a SPOT Advanced 4.6 software at 10× and 40× magnification. These micrographs were analyzed using Image J version 1.46 (Image Processing and Analysis in Java; US National Institutes of Health, Bethesda, MD, USA; https://imagej.nih.gov/ij/. (accessed on 12 April 2020)) by scanning 10 non-overlapping fields in each tissue and expressing positive areas as a percentage of the total area.

### 4.4. Reverse-Transcriptase qPCR

Total RNA (Ribonucleic Acid) was extracted from primary tumor samples using an RNeasy Mini Kit, (Qiagen; #74104, Venlo, Finland) according to the manufacturer’s protocol. A NanoDrop spectrophotometer (Thermo Fisher; MultiscanGo, MA, USA) was used to measure absorbance at A260/280 and A260/230 to assess RNA concentration and quantity. cDNA (complementary DNA) was obtained by reverse-transcriptase PCR method, according to the commercial kit protocol (EvoScript Universal cDNA Master Kit, Qiagen; #7912374001, Venlo, Finland). Gene expression analysis was performed with the obtained cDNAs by following the commercial kit protocol (FastStart Essential DNA Green Master, Qiagen; #6402712001, Venlo, Finland). The primers used in RT-PCR (real-time polymerase chain reaction) assays were designed with sequences based on references from the literature [53,54] (Table 2). The Ct (cycle threshold) value measured on the instrument by means of SYBR Green probe was averaged over 3 technical repetitions. All genes were normalized to a housekeeping gene (ribosomal 18S), and ΔΔCt (fold change (FC)) was calculated using the arithmetic formula for comparative quantification.

### 4.5. Bioinformatics Analysis

Microarray data of gene expression profiles from the GSE62598 dataset were downloaded from the Gene Expression Omnibus (GEO) (www.ncbi.nlm.nih.gov/geo/) (accessed on 23 June 2020) based on the GPL7202 platform (Agilent-014868 Whole Mouse Genome Microarray). Gene expression profiling in this dataset revealed distinct expression patterns associated with 4T1 subpopulations derived from breast tissues, primary breast tumors and liver and lung metastatic tumors. The dataset contained 3 tissue samples for each group. All data were processed using R software (www.r-project.org) (accessed on 25 June 2020). The limma package was used to identify DEGs between the primary/metastasis samples and breast tissue samples [55]. An adjusted *p* < 0.05 and an absolute log2 FC > 1 was considered statistically significant. To determine the potential biological processes and pathways of the overlapping DEGs, ingenuity pathway analysis (IPA, www.qiagen.com/ingenuity) (accessed on 03 July 2020) was performed, with *p* < 0.01 and absolute log2FC > 1 as the threshold values. Furthermore, RNA sequencing datasets and matched clinicopathological information of breast invasive carcinoma (BRCA) were downloaded from the TCGA database (https://tcga-data.nci.nih.gov/) (accessed on 15 August 2022). The mRNA expression of FoxO family members and SIRT1 were analyzed in TNBC and normal tissue overall survival by the Gene Expression Profiling Interactive Analysis Platform [56]. The expression levels of these genes were investigated in non-metastasis and metastasis situations of TNBC.

### 4.6. Statistical Analysis

Experiments were performed at least three times for each group. Statistical analyses were performed by variance with Dunnett’s post hoc test using Graph Pad Prism 8 software (San Diego, CA, USA). Data are shown as mean ± standard error of the mean and e considered statistically significant at * *p* ˂ 0.05.

## 5. Conclusions

Our findings show that SIRT1 could affect the target genes of FoxO proteins and that the tumor cell can fight to survive. We also showed that SIRT1 and FoxO proteins are associated with metastasis. The expression of SIRT1 and FoxO proteins is lower in primary tumors compared to metastatic areas, suggesting that SIRT1 and FoxO proteins trigger metastasis and that SIRT1 may interfere with the target genes of FoxO proteins in a particular pathway and affect tumor cell survival and metastasis potential.

## Figures and Tables

**Figure 1 ijms-23-10227-f001:**
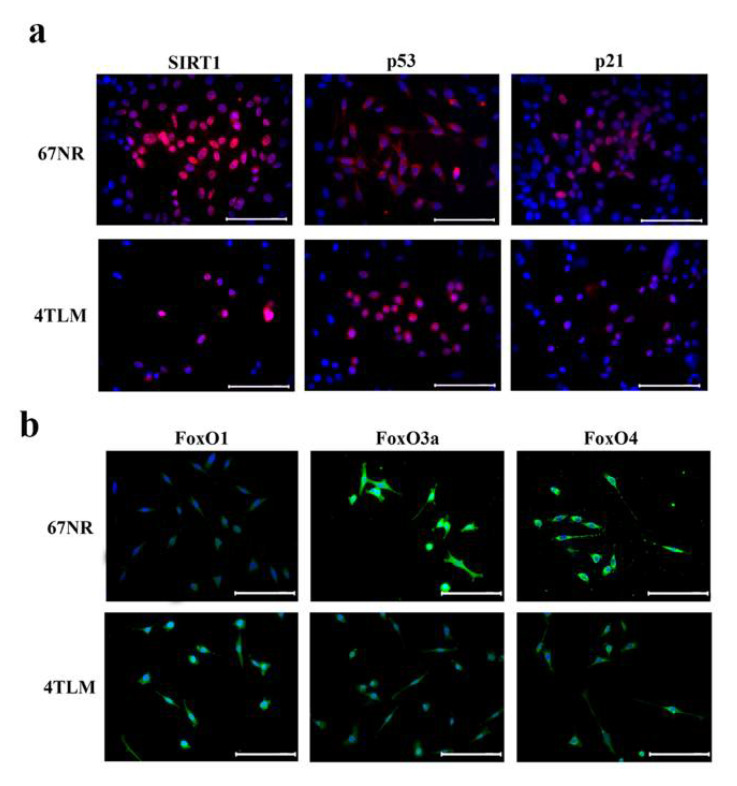
Immunofluorescence labelling of SIRT1, p53, p21 and FoxO proteins in 4TLM and 67NR cell lines. (**a**) Representative images showing the expression of SIRT1, p53 and p21 in 67NR and 4TLM cancer cells. Red signals represents target protein expression, and blue signals represent DAPI, which was used to stain the nucleus. (**b**) Representative images show in FoxO1, FoxO3a and FoxO4 expression in 67NR and 4TLM cancer cells. Green signals represent target protein expression, and blue signals represent DAPI, which was used to stain the nucleus. Scale bar represents 50 μm.

**Figure 2 ijms-23-10227-f002:**
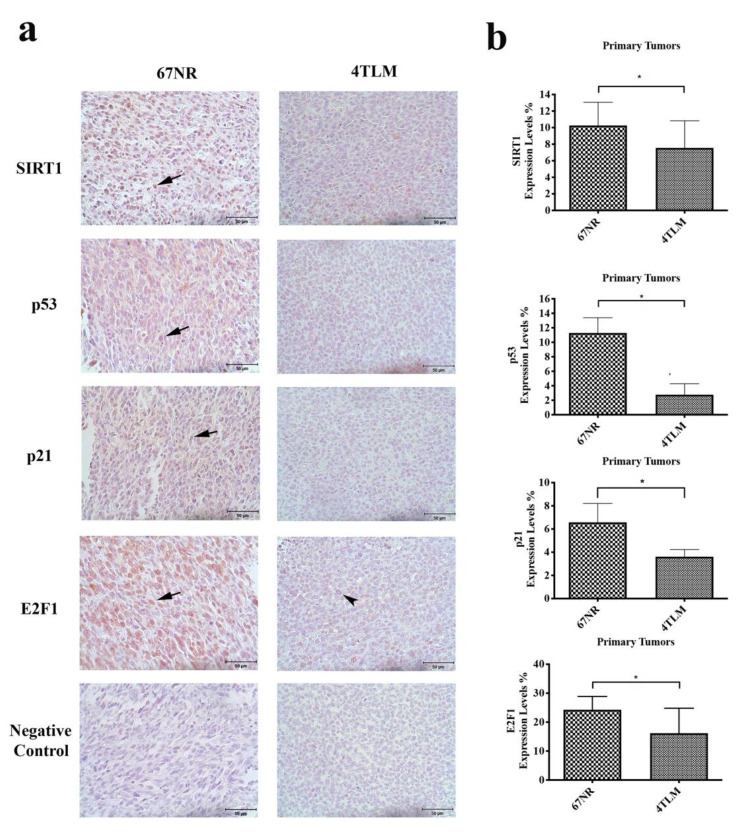
Expression of SIRT1, p53, p21, E2F1 and FoxO proteins, as well as cleaved caspase 3, in 67NR and 4TLM primary tumors. (**a**) Representative images showing the expression of SIRT1, p53, p21 and E2F1 in 67NR and 4TLM primary tumors. Arrows represent target protein expression for each protein, and the arrowhead represents perinuclear E2F1 expression in 4TLM. Scale bar represents 50 μm. (**b**) Graphs demonstrate the results of Image J analysis for each protein in primary tumor tissues (* *p* ˂ 0.05). (**c**) Representative images showing FoxO1, FoxO3a, FoxO4 and cleaved caspase 3 protein expression in 67NR and 4TLM primary tumors. Arrows represent target protein expression for each protein, and the arrowhead represents nuclear cleaved caspase3 expression in 4TLM. Scale bar represents 50 μm. (**d**) Graphs demonstrate results of Image J analysis for each protein in primary tumors (* *p* ˂ 0.05). (**e**) The graph represents mRNA levels of SIRT1, FoxO1, FoxO3a and FoxO4.

**Figure 3 ijms-23-10227-f003:**
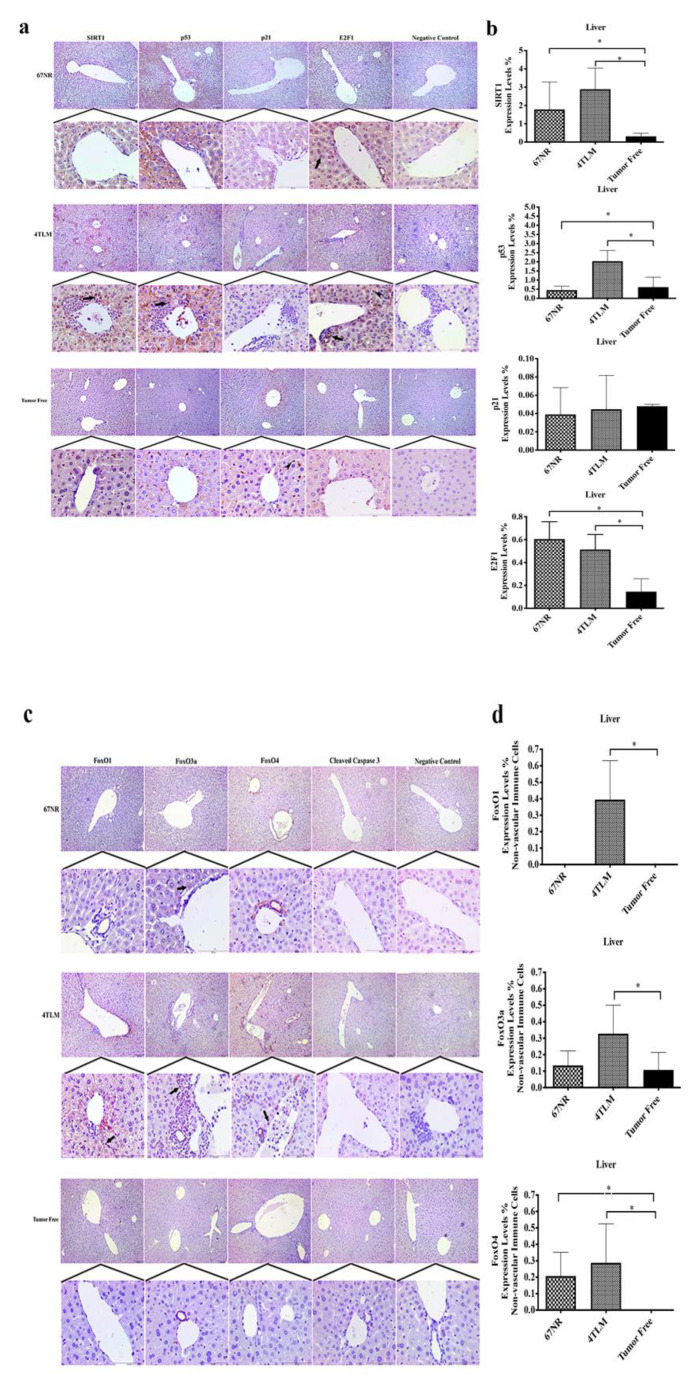
(**a**) Representative figures of SIRT1, p53, p21 and E2F1 immunohistochemical staining of liver tissues. Immunohistochemical reactions were interpreted compared to animals injected with both metastatic and non-metastatic cell lines, in addition to tumor-free animals. Arrows represent target protein expression for each protein in infiltrated cells, and arrowheads represent Kupffer cells. Scale bar represents 100 μm. (**b**) Graphs demonstrate the results of Image J analysis for each protein in liver tissues (* *p* ˂ 0.05). (**c**) Representative images of FoxO1, FoxO3a, FoxO4 and cleaved caspase 3 protein immunohistochemical staining of liver tissues. Arrows represent target protein expression for each protein in infiltrated cells. Scale bar represents 100 μm. (**d**) Graphs show the results of Image J analysis for each protein in liver tissues (* *p* ˂ 0.05).

**Figure 4 ijms-23-10227-f004:**
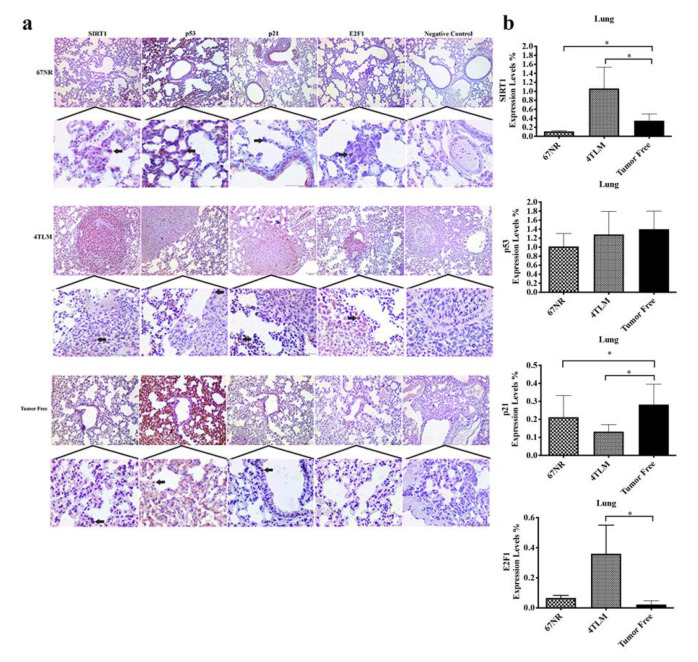
(**a**) Representative images of SIRT1, p53, p21 and E2F1 expression in lung tissue. Arrows represents target protein expressions for each protein in infiltrated cells. Scale bar represents 100 μm. (**b**) Graphs demonstrate the results of Image J analysis for each protein in lung tissues (* *p* ˂ 0.05). There was no significant difference in p53 expression. (**c**) Representative images of FoxO1, FoxO3a, FoxO4 and cleaved caspase 3 in lung tissue. Arrows represent target protein expressions for each protein in infiltrated cells. Scale bar represents 100 μm. (**d**) Graphs demonstrate the immunohistochemical results of Image J analysis for each FoxO protein in lung tissues (* *p* ˂ 0.05).

**Figure 5 ijms-23-10227-f005:**
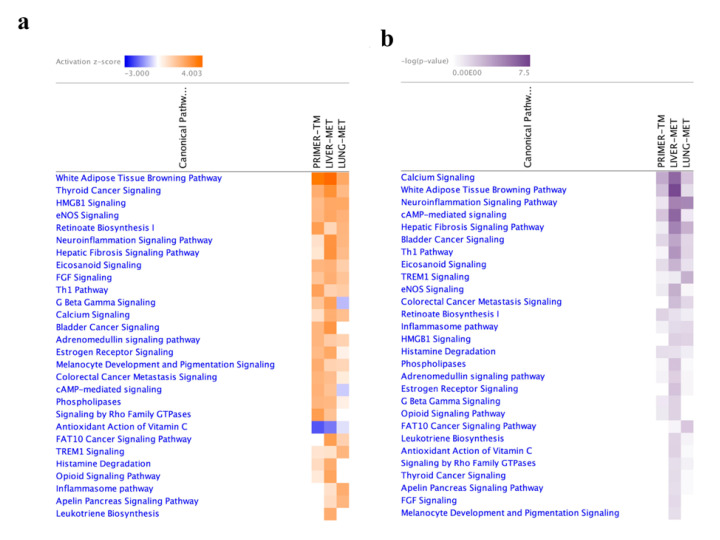
Canonical pathway comparison of 4T1 primary tumor and metastatic subpopulations by IPA with GSE62598 data. Heat maps show (**a**) activation score (absolute z-score > 2) and (**b**) *p* value absolute log *p*-value > 1.3. (**c**) Sirtuin pathway was generated from GSE62598 data that include the gene expression profile of 4T1 primary tumors and liver/lung metastatic subpopulations in a mouse model. The figure represents gene profiling in liver metastasis of 4T1 cells (absolute log *p*-value > 1.3). (**d**) Cell proliferation of tumor cells was found to have a regulator effect in liver and lung metastases of 4T1 cells by IPA. The intensity of the node color indicated the degree of regulation (red; upregulation, green; downregulation).

**Figure 6 ijms-23-10227-f006:**
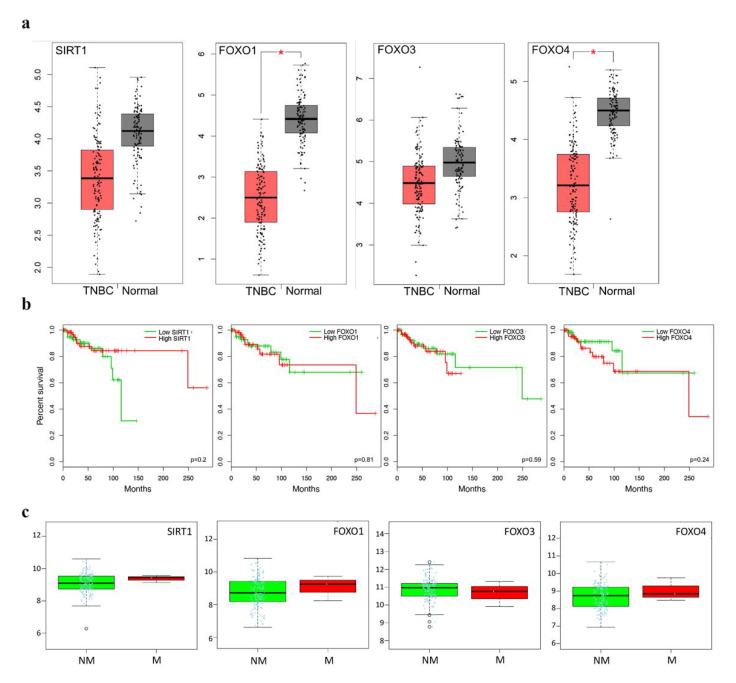
The mRNA expression of FoxO family members and SIRT1 in TNBC patients of TCGA. (**a**) mRNA expression of FoxO members and SIRT1 between TNBC and normal tissues in TCGA (*n*: 135, *n*: 112, respectively). * shows *p* value < 0.05. (**b**) Overall survival of TNBC patients according to Kaplan–Meier analysis (*n* = 134). (**c**) Comparisons of gene expression between tumor samples of TNBC patients in the metastasis stage (NM: non-metastasis, M: metastasis) (*n* = 137).

**Table 1 ijms-23-10227-t001:** Upstream regulators predicted to be activated or inhibited in liver and lung metastasis, with FDR *p* < 0.05, on the basis of known interactions compiled in the IPA (underlined gene names, targeted only in liver metastasis; italic gene names, targeted only in lung metastasis).

Upstream Regulator	Z Score	*p*-Value of Overlap	Target Genes
FOXO1 Liver-met Lung-met	0.92 1.427	1.19 × 10^−4^ 5.16 × 10^−7^	Acan, AFF3, ANGPT1, ANGPT2, APLN, APOA5, BLNK, BNIP3, CCL20, CCN2, CCN3, CCNG2, CD40, CERS4, CTSK, CXCL10, DAPK1, EDN1, EOMES, FABP4, FBXO32, *FLT4*, GADD45A, HSPD1, ICAM1, IFNG, IGF1R, IKZF2, *IKZF5*, ITGB2, ITGB6, JAG1, MMP9, OVOL1, PPARGC1A, PSMB8, PTGS1, RIPOR2, SELL, SEMA6D, SERPINB2, SERPINB5, SERPINE1, SESN3, SFN, SLC2A4, SOX5, TNFRSF9, TNFSF10, UCP1,V CAM, ACLY, ADIPOQ, AICDA, ATP6V0D2, BIRC3, CA2, CAMK4, CASP14, CCND1, CTSV, EDNRB, EFHD1, ERG, FASLG, FGF21, FOXC1, GABARAPL1, GPX3, IGLL1/IGLL5, IKZF1, IL1B, IL22, IL23R, IL7R, KLK3, LAMP2, MLXIPL, MYCN, PIK3C3, PNPLA2, POMC, PRKAA2, RUNX2, SCD, SOX9, TIE, *A2M*, *AQP9*, *BCL2*, *COL4A*, *EBF1*, *FLT4*, *GSTK*, *IKZF5*, *IL18*, *IL6*, *ITGAM*, *ITGAV*, *KIF11*, *MAP1LC3B*, *MRPL57*, *MYOG*, *PLA2G2D*, *PTEN*, *RICTOR*, *RPS6KA3*, *SREBF2*, *STAT5B*, *TBX21*, *TIMM8A*, *TKT*, *TNNC1*
FOXO3 Liver-met	0.82	2.94 × 10^−5^	Acan, ACLY, Acot1, ACTA2, ANGPT1, ANGPT2, APLN, AQP4, ATP6V0D2, BNIP3, CCN2, CCND1, CCND2, CCNG2, CDH1, CERS4, CLDN1, CTSV, CXCL10, EDNRB, ERG, FABP4, FASLG, FBXO32, FOXC1, GABARAPL1, GABRR2, GADD45A, GPX3, ICAM1, Ifna4, IFNG, IGF1R, IKZF2, IL12A, INHBA, ITGB2, JAG1, MAP1LC3A, MMP13, MMP9, Mt1, Mt2, NDRG1, NOS2, NUPR1, OVOL1, PLAU, POMC, PPARGC1A, PRKAA2, PTGS1, RTN3, RUNX2, SBSN, SELL, SERPINE1, SESN3, SLC40A1, SOX5, SOX9, TCIM, TIE1, TNFSF10, TP53INP1, TP63, VCAM1, VEGFA, VIM
FOXO4 Liver-met Lung-met	0.292 0.151	2.39 × 10^−8^ 1.0 × 10^−6^	Acan, ANGPT1, APLN, BNIP3, CCN2, CCNG2, CERS4, EDNRB, ERG, FABP4, FASLG, FOXC1, GADD45A, ITGB2, JAG1, MMP9, OVOL1, PTGS1, SELL, SERPINE1, SESN3, SOX5, TIE1, VCAM1, ACLY, CCND1, CCND2, GABARAPL1, GPX3, PRKAA2, PSMD11, RUNX2, SCD, SLC2A1, SOX9, VEGFA, *COL4A1*, *FLT4*, *IDI1*, *ITGAM*, *MAP1LC3B*, *PLA2G2D*, *RICTOR*, *SREBF2*
SIRT1 Liver-met Lung-met	1.011 −0.595	3.4 × 10^−4^ 3.27 × 10^−9^	ABCB1, APLN, BDNF, BNIP3, CDH1, DDAH2, FABP4, FBXO32, GADD45A, GBP3, GBP6, HLAA, HLADQB1, ICAM1, IFNG, IGF1R, Iigp1, LAMA4, LGALS3BP, MMP13, MMP9, PDGFRA, PNPLA3, PPARGC1A, PRDM16, RIMS2, RTP4, RUNX2, Sectm1b, SERPINE1, SLC7A11, SP110, SYNPO, TAC, STD2, TMPRSS4, ZEB1, ABCA1, ABCG1, ACAP1, ADIPOQ, AGT, BIRC3, Ccl2, CCND1, CCND2, CCNG2, CEBPB, CLEC10A, CTNNB1, EPAS1, FGF21, FGFR1, GABARAPL1, GLI2, GRIP1, GSTM3, Ifi47, IGHM, IL1B, KALRN, KMT2B, LAMA2, LRCH1, NANOG, NCAM2, NECTIN4, NKG7, NOS2, PCSK2, RBPJL, SLC27A6, TNFSF11, TP73, TRIM31, ZNF296, *AFP*, *BCL2*, *CCNG2*, *CD74*, *CMPK2*, *CNTN6*, *CPB*, *Csprs*, *CYP2B6*, *DDX60*, *DHX58*, *EDEMEP3*, *ERMP1*, *HIVEP3*, *IFI44*, *IFIT1B*, *IFIT3*, *Igtp*, *IL12B*, *IL6*, *IRGM*, *Irgm1*, *KDM5B*, *MYOG*, *NAIP*, *NF1*, *NLRC5*, *NTRK2*, *Oasl2*, *P3H3*, *PRDM1*, *PSMB9*, *TAP1*, *TFPI*, *Tgtp1/Tgtp2*, *TIMP2*, *Trim30a/Trim30d*, *UBA7*

**Table 2 ijms-23-10227-t002:** RT-PCR primer sequences.

	FORWARD	REVERSE
FoxO1	CTTCAAGGATAAGGGCGACA	GACAGATTGTGGCGAATTGA
FoxO3a	GCTAAGCAGGCCTCATCTCA	TTCCGTCAGTTTGAGGGTCT
FoxO4	TCATCAAGGTTCACAACGAGGC	AGGACAGACGGCTTCTTCTTGG
SIRT 1	TCGTGGAGACATTTTTAATCAGG	GCTTCATGATGGCAAGTGG
r18S (Housekeeping gene)	GGTGCATGGCCGTTCTTA	TCGTTCGTTATCGGAATTAACC

## Data Availability

The data that support the findings of this study are available from the corresponding author, G.T., upon reasonable request.

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
