# Peer review of "SIRT1/FOXO Signaling Pathway in Breast Cancer Progression and Metastasis"

_ijms, 2022, doi:10.3390/ijms231810227_

Round 1
Reviewer 1 Report
In the present study the major aim of the study is to evaluate the expressions of SIRT1 and FoxO proteins in metastatic and non-metastatic breast cancer cells and distant organs metastases. The study is well designed and please find my comments related to the methods and obtained results below,
Other than the SIRT1 and FoxO proteins, the p53, p21, and other proteins were not well described in the abstract, and their inclusion appears random. Please reorganize the abstract such that the significance is described first, followed by the findings section.
On the other hand, the explanation of the aforementioned proteins is excessive, resulting in a lengthy introduction that detracts from the study's emphasis. Authors are not obligated to cover every aspect of the proteins of interest in the introduction. Please also shorten the opening while maintaining its essential purpose of explaining the subject and making the research more accessible to the reader. It indicates that it is not necessary for each protein to be discussed in its own paragraph.
What is the cutoff used to pick differentially expressed proteins from a microarray dataset? Why was just a single dataset gathered for analysis? There are many more that are pertinent to the inquiry at hand. Why haven't writers used other datasets except this one?
To improve the specificity of the TCGA study, it would be preferable to disclose the survival plot and subset of the database used by the TCGA.
To provide better and more precise information, I strongly urge that writers utilize oncoplot, which will strengthen the overall argument.
Author Response
In the present study the major aim of the study is to evaluate the expressions of SIRT1 and FoxO proteins in metastatic and non-metastatic breast cancer cells and distant organs metastases. The study is well designed and please find my comments related to the methods and obtained results below
Reply: We really appreciate for your comments. We checked the entire manuscript and revised according to your suggestions. All revisions were marked using the "Track Changes" function.
Other than the SIRT1 and FoxO proteins, the p53, p21, and other proteins were not well described in the abstract, and their inclusion appears random. Please reorganize the abstract such that the significance is described first, followed by the findings section.
Reply: Thank you for your valuable comments. The abstract revised and added to the article by your suggestions.
On the other hand, the explanation of the aforementioned proteins is excessive, resulting in a lengthy introduction that detracts from the study's emphasis. Authors are not obligated to cover every aspect of the proteins of interest in the introduction. Please also shorten the opening while maintaining its essential purpose of explaining the subject and making the research more accessible to the reader. It indicates that it is not necessary for each protein to be discussed in its own paragraph.
Reply: We appreciate your comment. As you suggested, we have revised the introduction and with clearer information to make it pellucid and more understandable for the reader.
What is the cutoff used to pick differentially expressed proteins from a microarray dataset? Why was just a single dataset gathered for analysis? There are many more that are pertinent to the inquiry at hand. Why haven't writers used other datasets except this one?
Reply: The cutoff for differentially expressed proteins is FDR-adjusted P value <0.05 and absolute log2 fold change of >1 and was added in manuscript. The cutoff values used in IPA were presented in manuscript.
We searched a large part of the GEO dataset (over 100) online. Our keywords for this research are mouse model, gene expression, 4T1 breast cancer, liver and lung metastases. In addition, we wanted this data to be a published data because it is important for a journal or reviewers to review it for the safety of this data. We obtained some data in table 1. Among them, we selected GSE62598 because it is the only data that reflects our study, including both liver and lung metastases, and published before.
Tablo 1: Data set obtained from GEO
|
GEO number |
Features of data |
Published |
|
GSE62598 |
4T1 breast cancer, gene expression, mouse model, and brain, liver, and lung metastases |
Yes |
|
GSE54773 |
4T1 breast cancer, gene expression, mouse model, and brain and lung metastases |
Not |
|
GSE149635 |
4T1 breast cancer, gene expression, mouse model, and lung metastasis |
Not |
|
GSE150928 |
4T1 breast cancer, gene expression, mouse model, and lung metastasis |
Not |
|
GSE146012 |
4T1 breast cancer, gene expression, mouse model, and lung metastasis |
Yes |
|
GSE110101 |
4T1 breast cancer, gene expression, mouse model, and brain and lung metastases |
Yes |
|
GSE104264 |
4T1 breast cancer, gene expression, mouse model, and lung metastasis |
Yes |
To improve the specificity of the TCGA study, it would be preferable to disclose the survival plot and subset of the database used by the TCGA.
Reply: We added the survival plots of TNBC patients from TCGA in figure 6, according to reviewer’s suggestion.
To provide better and more precise information, I strongly urge that writers utilize oncoplot, which will strengthen the overall argument.
Reply: Thank you for your valuable comments. In this article, we did not include oncoplot because we did not discuss a subject related to any mutation and gave some information with graphics.

Reviewer 2 Report
The manuscript is interesting and well written. The provided experiments are accurate and the observed results are reasonable and support the conclusions. There are some stylistic issues the authors should deal with in a revised version in order to improve the manuscript:
Abstract: The following sentence is incomplete: ´´In our study, in vitro evaluation of SIRT1, p53, p21, and FoxO proteins 24 using metastatic 4TLM and 67NR cell lines.´´
Figures 2-5: The readability of these figures must be improved. In the current form, the text and data in these figures is too small.
Table 1 looks like being pasted from a graphics file. Maybe the authors better use the table form of the IJMS manuscript template.
Author Response
The manuscript is interesting and well written. The provided experiments are accurate and the observed results are reasonable and support the conclusions. There are some stylistic issues the authors should deal with in a revised version in order to improve the manuscript:
Reply: Thank you for your valuable comments. We checked the entire manuscript and revised according to your suggestions. All revisions were marked using the "Track Changes" function.
Abstract: The following sentence is incomplete: ´´In our study, in vitro evaluation of SIRT1, p53, p21, and FoxO proteins 24 using metastatic 4TLM and 67NR cell lines.´´
Reply: Thank you for reminding us for this mistake. We had rewritten the sentence as “In our study, in vitro evaluation of these proteins was performed using metastatic 4TLM and 67NR cell lines.”
Figures 2-5: The readability of these figures must be improved. In the current form, the text and data in these figures is too small.
Reply: Thank you for your valuable attention and comments. Figure 2-5 renewed by your suggestions. Figure letter, axis and axis captions are enlarged and revised.
Table 1 looks like being pasted from a graphics file. Maybe the authors better use the table form of the IJMS manuscript template.
Reply: We appreciate for your comments. We revised tables using the table form of the IJMS manuscript template.

Round 2
Reviewer 1 Report
no more comments